# Postoperative C-Reactive Protein Trend Is a More Accurate Predictor of Anastomotic Leak than Absolute Values Alone

**DOI:** 10.3390/jcm14092931

**Published:** 2025-04-24

**Authors:** Britney R. Niemann, Jeevan Murthy, Connor Breinholt, Jacob Swords, Alyson Stevens, Mary Garland-Kledzik, Keri Mayers, Emily Groves, Kevin Train, Douglas Murken

**Affiliations:** 1Department of Surgery, School of Medicine, West Virginia University, Morgantown, WV 26506, USA; brharris1@hsc.wvu.edu (B.R.N.); jjm0030@mix.wvu.edu (J.M.); connor.breinholt@hsc.wvu.edu (C.B.); jacob.swords@hsc.wvu.edu (J.S.); molly.kledzik@hsc.wvu.edu (M.G.-K.); keri.mayers@hsc.wvu.edu (K.M.); emily.groves@hsc.wvu.edu (E.G.); kevin.train@hsc.wvu.edu (K.T.); 2Cancer Institute, West Virginia University, Morgantown, WV 26506, USA; ams0021@mix.wvu.edu; 3Department of Microbiology, Immunology and Cell Biology, West Virginia University, Morgantown, WV 26506, USA

**Keywords:** anastomotic leak, C-reactive protein, colorectal surgery

## Abstract

**Background/Objectives**: An anastomotic leak (AL) following colorectal surgery is one of the most feared complications due to its associated morbidity and mortality. Early detection of ALs remains difficult, as the development of clinical signs of deterioration can be a late finding. This is particularly problematic in patients with poor access to care after discharge. C-reactive protein (CRP) is a systemic marker of inflammation that has been proposed as an early AL screening. However, absolute cut-off values have been shown to have limited sensitivity and specificity. We propose the use of CRP trends for early AL detection. **Methods**: A retrospective chart review of patients undergoing surgery requiring at least one anastomosis at a single tertiary care center was performed. Patients with two or fewer postoperative CRP values were excluded. Postoperative CRP trends were compared between control and AL patients using a mixed model with a Geisser–Greenhouse correction. **Results**: CRP trends differed significantly between AL and control patients, with a 10% CRP increase after postoperative day two showing 100% sensitivity and 84% specificity for an AL as well as a 100% negative predictive value. Accepted CRP cut-off values on postoperative days three and four had sensitivities of only 71.4% and 80% and specificities of 70.0% and 76.5%, respectively. CRP trends differed in AL versus control patients despite the surgical approach or presence of additional procedures. **Conclusions**: Daily monitoring of CRP trends (versus absolute cut-offs) may enhance early anastomotic leak detection and aid in discharge decision-making, particularly important in rural settings with limited healthcare access.

## 1. Introduction

An anastomotic leak (AL) following colorectal surgery is one of the most feared complications due to its high associated morbidity and mortality. The incidence of AL varies between 2.8 and 30% [1,2,3]. This wide variability is the result of multiple patient characteristics and risk factors, including age, malnutrition, steroid use, diabetes, tobacco use, and cardiovascular disease, as well as clinical factors such as the primary disease state and the location of the anastomosis [2,4]. Furthermore, the definition of leak varies by study. In addition to prolonging hospital stays, AL may also impact oncologic outcomes [3,5,6,7,8,9].

With the implementation of enhanced recovery after surgery (ERAS) pathways, patients are being discharged earlier [10,11,12,13]. This makes early detection of complications important, especially in patients with limited postoperative access to care after discharge. One challenging aspect of AL management is the delayed presentation. Many patients present 6–8 days postoperatively and even as far out as 30 days after surgery [14,15]. Early detection of AL is associated with higher anastomotic salvage rates and less severe complications, as defined by the Clavien–Dindo classification system [16]. Oncologic outcomes may also be better with early detection. Gozalichvili et al. found that patients with an early AL diagnosis trended toward increased adjuvant therapy completion (18.9% vs. 39.5%; *p* = 0.058) and decreased mortality (1.4% vs. 9.8%; *p* = 0.081); however, neither of these data points was statistically significant [16].

For this reason, many authors have attempted to define methods for the early detection of AL. DIACOLE is a 13-point scoring system consisting of signs and symptoms of an AL [17]. The score components are weighted differently, and thus, the authors have developed software to perform calculations. DIACOLE was found to have good predictive power, but the complicated score calculation can be cumbersome in a clinical setting. Other authors have investigated the use of peritoneal fluid markers in AL diagnosis, including bacteria, interleukin-6, and tumor necrosis factor alpha [18,19,20]. However, ERAS pathways discourage the routine use of intraabdominal drains, making drain sampling for AL detection less ideal [21].

C-reactive protein (CRP) is an acute-phase reactant produced and released by the liver in the setting of acute inflammation, such as after surgery [22]. It typically exists only at low levels in healthy individuals. The levels of other acute-phase reactants, including procalcitonin and interleukin-6 (IL-6), are well known to increase in the serum when responding to a pro-inflammatory environment. Several studies have attempted to validate their utility as detectors of early AL, yet results are inconclusive. Baseline CRP values are impacted by chronic inflammatory conditions such as rheumatoid arthritis and lupus, as well as other patient characteristics like age, gender, smoking status, and BMI [22,23,24]. Interindividual variability independent of these factors has also been described [22,25,26]. Prior studies have used a range of absolute CRP thresholds to diagnose postoperative AL with poor sensitivity and specificity [27].

In this institutional retrospective study, we sought to confirm the utility of an absolute CRP threshold in the diagnosis of AL, as well as examine the role of day-to-day trends. The decision to focus on CRP trends, rather than relying solely on absolute values, was based on the knowledge that CRP levels exhibit significant baseline variability, as described above. By examining trends over time, we aimed to mitigate the potential confounding effects of baseline CRP variability and provide a more accurate and nuanced marker for the early detection of AL.

## 2. Materials and Methods

### 2.1. Study Sample

After Institutional Review Board approval was obtained, patients over the age of 18 who underwent elective colorectal surgery from January 2021 to February 2023 were retrospectively reviewed. AL rates within 90 days of operation were examined. AL was defined as enteric contents in the peritoneal cavity at the time of re-operation or in the patient’s drain (either placed during the index operation or postoperatively during interventional radiology). Patients who developed an AL continued to be collected through May 2024 in order to increase the sample size of this cohort. Patients were excluded from the study if they had fewer than three postoperative CRP values or underwent emergency surgery.

### 2.2. Data Collection

Patients who did not develop an anastomotic leak constituted the control group. Age, sex, comorbidities, and surgical details were retrospectively collected for all patients. Postoperative CRP and WBC levels were collected. The number of days that CRP levels were recorded was provider dependent. The presence of conditions that can elevate the postoperative CRP, such as a concomitant infection (pneumonia or urinary tract infection (UTI), was also recorded.

### 2.3. Statistical Analysis

Continuous variables were tested for normality, with non-parametric variables reported using the median and interquartile range (IQR). A Mann–Whitney test was used for non-parametric data. Categorical variables were reported as percentages with frequencies and analyzed using a Fisher’s exact test or Chi-square. We could not analyze postoperative CRP and WBC trends with a repeated measures ANOVA due to missing values. Therefore, we analyzed the data by fitting a mixed model, as implemented in Prism Version 10.2.3 (GraphPad Software, LLC, Boston, MA, USA). A Geisser–Greenhouse correction was performed.

## 3. Results

Two hundred and fifty-six colorectal patients with available postoperative CRP values were retrospectively identified. Of this group, 29 patients had two or fewer postoperative CRP values and were excluded. Of the remaining 226 patients, six (2.7%) developed an AL. Given the small number of patients with AL, we continued to collect AL patients until a sample size of nine was reached. The median day of leak diagnosis was postoperative day four. Patient characteristics are shown in Table 1. There was no difference between patients who developed an AL versus controls with respect to comorbidities associated with elevated CRP levels, including CKD requiring dialysis, liver disease, inflammatory bowel disease, or BMI. Similarly, we quantified the incidence of concomitant postoperative infectious complications unrelated to the primary pathology, such as pneumonia or a UTI, which could impact the CRP level. Equal infection rates were seen in control and AL patients (13.2% vs. 11.1%, respectively; *p* = 0.99). Malignancy was the most common surgical indication in the control group (37.3%) versus benign disease in the AL group (44.4%) (Table 1).

However, there was no significant difference in the distribution of surgical indications between groups. The median number of anastomoses performed was one in each group, with the maximum being three in the control group and two in the AL group (*p* = 0.94). Patients who received more than one anastomosis were commonly patients with fistulas requiring multiple resections at the index operation. The types of anastomoses performed are shown in Table 1, with the AL group having more distal anastomoses overall. A diverting ostomy was created in 9.6% of control patients but none of the nine AL patients (*p* = 0.99) (Table 1).

We first evaluated the use of standard CRP level cut-offs for AL screening in our patients (Table 2). CRP over 160 mg/L on postoperative day three had a sensitivity of 71.4% and a specificity of 70.0%. CRP over 150 mg/L on postoperative day four had a sensitivity of 80% and a specificity of 76.5%. These results are similar to those reported in prior studies [27]. We observed significant variation in CRP levels between patients who had similar operations. For example, the maximum postoperative CRP in control patients following a local or laparoscopic-assisted ileostomy closure ranged from 5.3 to 305.6 mg/L (Figure 1A).

Given this significant interpatient variability, we hypothesized that the day-to-day trend in postoperative CRPs, as opposed to absolute values, would be more predictive of AL. Postoperative CRP trends were significantly different between AL and control patients (Figure 1B,C). Interestingly, WBC trends, another marker of acute inflammation and infection, did not differ between control and AL patients (Figure 1D). Initial CRP decrease was found in control patients between postoperative days two to seven, with the majority decreasing on postoperative day three. The range of CRP reduction from the maximum on the first day of decline in control patients is shown in Figure 1E, with the median being 27.2%. After postoperative day two, an increase in CRP by 10% or more between any two consecutive postoperative days had a sensitivity of 100% and specificity of 83.6% for an anastomotic leak (Table 2). The negative predictive value (NPV) was 100%. Similarly, we examined the use of a 10% decline in CRP level between any two consecutive postoperative days following postoperative day two. The sensitivity was much lower (75%) with similar specificity (88.6%) and NPV (99%) (Table 2).

With minimally invasive approaches becoming increasingly common in colorectal surgery, we also evaluated the effect of surgical approach on postoperative CRP trends. Minimally invasive approaches are thought to decrease the inflammatory response following surgery and could, therefore, impact CRP levels. About half of the control patients underwent a minimally invasive approach compared to a third of the AL patients (48.2% vs. 33.3%, respectively; *p* = 0.50). A significant difference in postoperative CRP trends was found in control patients who underwent open versus minimally invasive surgery (Figure 2A). There was no difference in AL patients based on operative approach. However, these groups are difficult to compare as there were only three AL patients who underwent minimally invasive surgery, and their leaks were all diagnosed by postoperative day three. Differences in the postoperative CRP trend persisted when comparing control and AL patients despite surgical approach, although the difference was more dramatic in patients undergoing open surgery (Figure 2C,D).

Some patients also underwent a second procedure during the index operation. For example, a colectomy for colon cancer may have been performed with a resection of liver metastases. Similar to the differences in inflammatory response between minimally invasive and open surgical approaches, we hypothesized that additional procedures could impact the postoperative inflammatory process. Five (55.6%) AL patients and 66 (30.0%) control patients underwent a second procedure during the index operation. Additional procedures are listed in Appendix A. A significant difference in postoperative CRP trends was found in control patients undergoing colectomy alone compared to those who also underwent concomitant procedures (Figure 3A). This difference was not seen in AL patients (Figure 3B). We again found differences in CRP trends between control and AL patients, regardless of additional procedures being performed (Figure 3C,D).

## 4. Discussion

An anastomotic leak following surgery can have a profound impact on healthcare costs and patient outcomes [28]. Not only does an AL result in longer length of stay with frequent need for reoperation, but it may also impact oncological outcomes [29,30,31]. We postulate that the oncological impact is secondary to hospitalizations or complications delaying the initiation of adjuvant therapy. Several studies have demonstrated decreased long-term colorectal cancer survival rates when adjuvant therapy is initiated more than eight weeks from the index operation [32]. Consistent with this theory, Gozalichvili and colleagues demonstrated that early AL diagnosis was associated with less invasive surgical management, shorter length of stay, and decreased surgical complications with a trend toward increased adjuvant therapy completion [16].

To date, CRP cut-off values on specific postoperative days have been used to screen for AL in hopes of enabling early diagnosis. A wide variety of thresholds have been reported, ranging from 123 to 209 on postoperative day three and 56 to 180 on postoperative day four [33,34,35]. However, given the number of factors influencing CRP levels, it is not surprising that the same CRP threshold yields different results depending on the study [2,4,33,34,35,36]. We found that previously reported absolute CRP cut-offs had limited sensitivity and specificity for AL in our patient population.

Since there are wide ranges of CRP responses to surgery depending on both patient factors and surgical intervention, it follows that using an individual patient’s daily CRP trend may provide more accurate insight into the evolving inflammatory process. For instance, the PREDICT study found a CRP increase of over 50 mg/L between postoperative days to be highly specific for AL [35]. However, these findings lacked sensitivity with changes between postoperative days two to five, having a maximum sensitivity of only 32%. When the authors investigated the utility of this CRP change between any two postoperative days, the sensitivity increased to 85% at the cost of specificity (51%). Other acute-phase reactants, such as IL-6 and procalcitonin, have also been investigated with mixed results. These studies have also used cut-off values on particular postoperative days, resulting in a large tradeoff between sensitivity and specificity. For example, El Zaher and colleagues found that procalcitonin could reach a specificity of 96.7% on postoperative day five, but the sensitivity was only 77.3% [37]. In another study by Zielińska-Borkowska, the authors found less impressive predictive value with procalcitonin and reported that IL-6 was not a good marker for AL [38]. Given that these values are not routinely obtained postoperatively in our patients, we did not include them in this retrospective study. Interestingly, WBC trends did not differ significantly between controls and AL patients. This may be, in part, due to less dramatic increases in WBC shortly after the physiologic de-margination response after surgery. Additionally, we hypothesize that the short half-life of CRP allows for rapid rises and falls in levels to better reflect acute inflammatory changes [22,25].

The ideal screening test would, of course, have both high sensitivity and specificity. As seen in the PREDICT study, there is often a trade-off between the two. The PREDICT study highlights an important point in that a CRP trend may be more useful than absolute cut-offs in predicting AL. However, the authors still incorporate the numeric value to a certain extent, as the trend is defined by a difference of 50 mg/L between postoperative days. In this study, we instead used a percent change in CRP as this may partially control for confounding factors influencing baseline CRP levels, such as obesity or the individual response to the surgical insult. We found that an increase by at least 10% on any two consecutive days after postoperative day two was 100% sensitive for AL and 83.6% specific, with an NPV of 100. This demonstrates the increased utility of CRP trends over absolute cut-off values.

Minimally invasive approaches to surgery are becoming increasingly common and are thought to produce a diminished inflammatory response when compared to open operations. The use of CRP to predict AL in this setting, compared to more invasive operations, has been poorly examined. One meta-analysis by Paradis et al. compared postoperative CRPs between laparoscopic and open colorectal surgeries in relation to all infectious postoperative complications, including AL [39]. Lower CRP thresholds were used to investigate laparoscopic surgery and had similar diagnostic characteristics to open surgery with higher CRP cut-offs. This demonstrated that CRP could be utilized in both settings, but that, again, a single CRP threshold could not be applied to all surgical patients. Similarly, it is reasonable to assume that the degree of inflammatory response would also differ based on the number of procedures performed, further complicating the use of absolute CRP cut-offs.

We therefore investigated the utility of CRP trends in the context of surgical factors. Control patients were found to have a more dramatic and prolonged inflammatory response after an open approach or multiple procedures. Their CRP trends differed significantly from the other control patients, whose CRPs plateaued starting on postoperative day two. In contrast, AL patients had similar postoperative trends regardless of these surgical factors. Despite the underlying variability in control patients’ CRPs, a significant difference in postoperative CRP trends persisted between AL and control patients. These findings support the use of CRP trends as opposed to absolute values.

As hospital stays shorten with the use of ERAS protocols, early identification of potentially life-threatening complications, such as an AL, is of the utmost importance. This is particularly true for rural patient populations such as ours. Patients often live hours from our facility with limited access to care after discharge. Therefore, early detection of AL prior to discharge is essential to ensure that these patients receive timely care. With high sensitivity and specificity, an upward-trending CRP should alert clinicians to a potential AL and prompt additional monitoring or testing even if the patient lacks clinical signs of an AL, including leukocytosis and fever. Conversely, given the predictive value of postoperative CRP trends demonstrated in this study, a down-trending CRP may provide clinicians with reassurance that their patients are safe for discharge.

There are several limitations to this study. First, our AL patient population was small. Given the low incidence of anastomotic leaks, patient accrual is slow. While our data show a significant difference in postoperative CRP trends, future multicenter studies, which would allow for higher sample sizes in a more timely manner, are needed to validate our findings in a larger population. Second, only elective operations were included in this study. Therefore, no conclusions can be made regarding the use of CRP trends in urgent or emergent operations. Third, during this data collection period, we had no department-wide CRP protocol to guide collection or clinical action based on results. While most providers ordered daily postoperative CRPs, some patients had intermittent missing values. Furthermore, the decision to complete additional workup for an AL was based on each provider’s clinical decision-making and not a particular CRP trend. As previously noted, we defined an AL as enteric contents in the peritoneal cavity at the time of re-operation or in the patient’s drain (either placed during the index operation or postoperatively during interventional radiology). Therefore, patients with perianastomotic air foci, phlegmon, or abscesses on imaging were not necessarily included unless drain placement revealed enteric contents. These findings may have represented micro-perforations. We decided to exclude them, however, as the inflammatory response, and thus CRP trend, would be expected to differ from a more clinically significant leak. Prospective trials should include a standardized protocol prompting additional imaging with the inclusion of these patient populations. Prospective trials with a standardized protocol prompting additional imaging based on CRP trends are needed to determine if CRP trends lead to earlier AL diagnosis. Another possible shortcoming of this study was our inability to evaluate the utility of other acute-phase reactants, including procalcitonin and IL-6, as methods in the detection of early AL. Lastly, we did not examine the cost of trending daily CRPs in patients.

In conclusion, postoperative CRP trends, rather than absolute values, are a more sensitive and specific method by which to screen for AL. We demonstrate the utility of CRP to predict AL in the setting of minimally invasive and open surgeries, as well as operations performed in conjunction with additional procedures. Based on these findings, we would recommend trending daily postoperative CRPs with a rise in CRP by 10% serving as a prompt for additional imaging to investigate for an anastomotic leak. However, further investigation is needed to confirm these findings in larger patient populations.

## Figures and Tables

**Figure 1 jcm-14-02931-f001:**
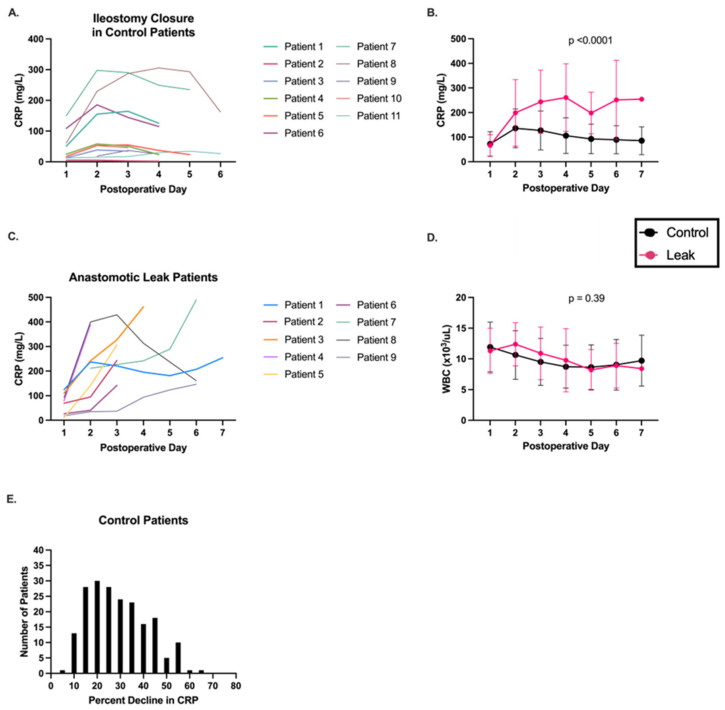
Postoperative CRP trends differ in patients with anastomotic leaks compared to controls. (**A**) The variability in absolute CRP levels is demonstrated by postoperative CRP trends in control patients who underwent the same operation, an ileostomy closure. (**B**) Patients with an AL demonstrate distinct postoperative CRP trends compared to control patients without AL. (**C**) The CRP trends of the nine AL patients until the day AL was diagnosed are shown, with most patients demonstrating a consistent uptrend. (**D**) Postoperative white blood cell trends were similar between control and AL patients. (**E**) The degree to which the CRP decreased on the first day of decline varied widely in control patients, with a median of 27.2%.

**Figure 2 jcm-14-02931-f002:**
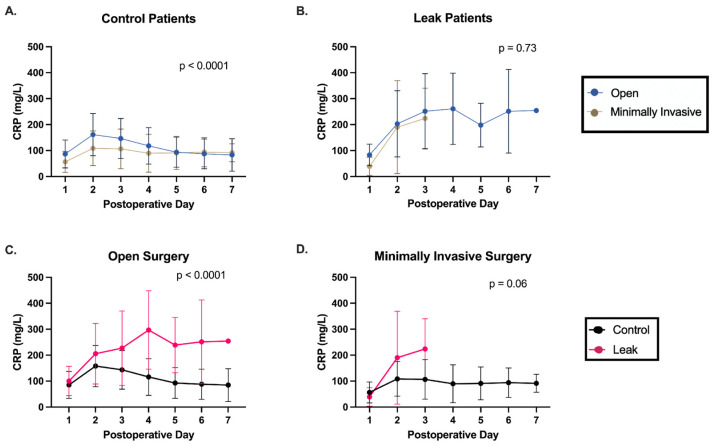
Differences in postoperative CRP trends, persistent despite surgical approach. (**A**,**B**) Control patients demonstrate a more robust inflammatory response after open surgery compared to minimally invasive surgery, whereas AL patients have similar CRP trends regardless of operative approach. (**C**,**D**) Comparison of postoperative CRP trends after open surgery and minimally invasive surgery shows that patients with an anastomotic leak have a distinct postoperative trend compared to control patients.

**Figure 3 jcm-14-02931-f003:**
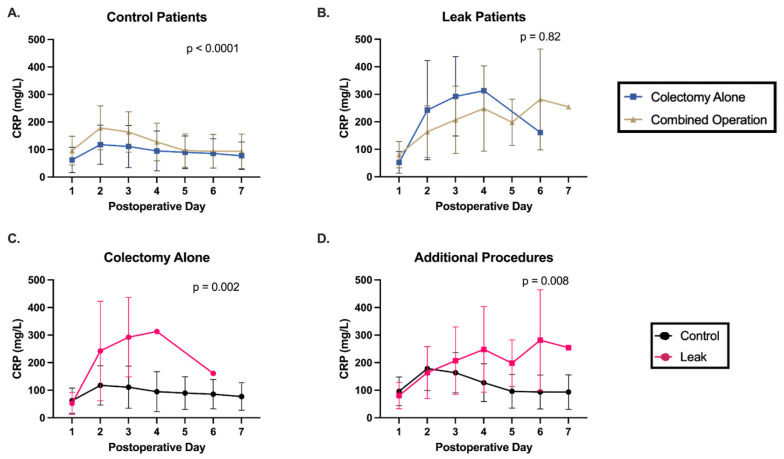
Postoperative CRP trends differ and combined operations influence postoperative CRP trends. (**A**) Control patients undergoing colectomy alone had a blunted inflammatory response compared to control patients who underwent a combined operation. (**B**) This difference was not seen within the AL patient group. (**C**,**D**) There is a significant difference in postoperative CRP trends between control and AL patients in the setting of colectomy alone as well as multiple operations.

**Table 1 jcm-14-02931-t001:** Patient characteristics and management.

	Control *n* = 220	Anastomotic Leak *n* = 9	*p*
Age, median years (IQR)	59.0 (46.0–69.0)	56.0 (44.5–67.0)	0.50
Female, *n* (%)	128 (58.2)	5 (58.2)	0.99
BMI, median (IQR)	28.6 (24.8–33.1)	30.1 (25.4–37.7)	0.31
History of CKD Requiring Dialysis, *n* (%)	5.0 (2.3)	0 (0)	0.99
History of Liver Disease, *n* (%)	10.0 (4.8)	0 (0)	0.99
History of Inflammatory Bowel Disease, *n* (%)	49.0 (98.0)	1.0 (11.1)	0.69
Active Tobacco Use, *n* (%)	62 (28.2)	4 (44.4)	0.28
Concomitant Operations, *n* (%)	98 (44.5)	6 (66.6)	0.31
**Surgical Indication**, *n* (%)			0.31
Elective stoma reversal	34 (15.5)	3 (33.3)	
Benign	62 (28.2)	4 (44.4)	
Malignant	82 (37.3)	2 (22.2)	
Inflammatory Bowel Disease	37 (16.8)	0 (0)	
Other	5 (2.3)	0 (0)	
**Types of Anastomoses**, *n* (%)			0.0001
Small Bowel–Small Bowel	30.0 (13.6)	1.0 (11.1)	
Ileocolic	100.0 (45.5)	2.0 (22.2)	
Ileorectal	4.0 (1.8)	0.0 (0.0)	
Colo-colonic	14.0 (6.4)	2.0 (22.2)	
Mid-to-High Colorectal	76.0 (34.5)	3.0 (33.3)	
Low Colorectal	20.0 (9.1)	1.0 (11.1)	
Ileoanal	4.0 (1.8)	0.0 (0.0)	
Coloanal	0.0 (0.0)	1.0 (11.1)	
Diverting Ostomy, *n* (%)	21 (9.6)	0.0 (0.0)	0.99

**Table 2 jcm-14-02931-t002:** Diagnostic characteristics of CRP cut-off values and trends.

Laboratory Finding	Sensitivity	Specificity	Positive Predictive Value	Negative Predictive Value
CRP > 160 on POD 3	71.4	70.0	7.0	98.7
CRP > 150 on POD 4	80.0	76.5	9.1	99.2
10% CRP increase *	100.0	83.6	18.2	100.0
10% CRP decrease *	75.0	88.6	19.4	99.0

Abbreviations: C-reactive protein (CRP); postoperative day (POD). * Between any postoperative days after POD 2 (i.e., POD 2 to 3, POD 3 to 4, POD 4 to 5, etc.).

## Data Availability

The raw data supporting the conclusions of this article will be made available by the authors upon request.

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
