# Peer review of "Postoperative C-Reactive Protein Trend Is a More Accurate Predictor of Anastomotic Leak than Absolute Values Alone"

_jcm, 2025, doi:10.3390/jcm14092931_

Round 1

Reviewer 1 Report

Comments and Suggestions for Authors

Niemann et al performed a study evaluating postoperative trends in CRP to detect complications early. The manuscript is well-organized and easy to follow. The novel approach of using CRP trends, rather than absolute values, to predict anastomotic leak is a potentially practice-changing finding. However, the manuscript can often be difficult to follow as the narrative is not clear. The authors present a significant amount of data without focusing on developing a clear story.

Major revisions:

  • The introduction includes relevant research, but would benefit from explicitly describing how early detection of anastomotic leak can reduce overall morbidity, improve adjuvant chemotherapy adherence, etc. For example, the reference to the study by Gozalichvili (misspelled in manuscript) et al should describe how early detection of anastomotic leak led to greater adherence to adjuvant therapy (delayed discharge, early imaging and detection, etc)
  • It would improve the narrative of the study for the authors to describe why trend in CRP was chosen instead of absolute value. This question is not addressed until the results section, where it seems the authors formed their hypothesis.
  • It seems that the control would be more robust to include all patients that did not develop leak. Is there a reason why the authors did not keep patients for 3 total CRP values, or to have them send off a CRP value at home after discharge for a total of 3 values?
  • This study only had 9 cases out of 226 patients. Do the authors believe this is enough to have a statistically significant and meaningful report? If so, can the authors please expound on how 9 cases are sufficient to describe an impactful finding? For example, it is very surprising to see that none of the patients who leaked had an operation for IBD, patients who are known to be in a chronically inflammatory state and likely on immunosuppression. It would therefore be difficult to extrapolate these findings as patients with IBD are high. The authors address the low case number in the limitations, but the authors need to include how statistical conclusions can still be drawn from such a small number.
  • How do the authors account for patients who might have leaked after discharge and presented to an outside hospital?
  • Can the authors explain why nutritional status or index versus repeat operation were not included as independent variables? Repeat operations have been found to be associated with higher risk of leak among patients with IBD
  • How was the decision made for re-operation on these patients?

Minor revisions:

  • Can the authors include diverting ostomy as a variable in Table 1?
  • The graphs in Figure 1 are very confusing. There is a grey rectangle without explanation under graph B. The caption is difficult to follow with the reference letter at the end of the sentence instead of the beginning. One of the case patients has a significant decline in CRP. E references the percent decline of CRP on the first day, which is not clear. Was there a reference CRP done pre-op, or is this the percent decline from POD1 to POD2? When were labs drawn?
  • In Table 2, can the authors clarify within the table itself? Making the language more straightforward will help the readers follow along. For example, 10% increase in CRP from POD2 to POD3
  • Why does anastomotic leak not include peri-anastomotic abscess that warrants a drain or antibiotics, similar to Gozalichvili's study? The author's definition only includes those who require re-operation
  • Instead of table 3, could the authors include blood loss, OR time, and whether a second procedure was performed in Table 1? Would be more clinically relevant.
  • According to Figure 2, it seems that patients undergoing minimally invasive surgery are more likely to leak earlier in their post op course than those undergoing open surgery. Is this true?
  • Can the authors recommend a post-operative protocol for surgical teams to follow for early detection of anastomotic leak after elective colorectal surgery?

Author Response

Reviewer #1

Niemann et al performed a study evaluating postoperative trends in CRP to detect complications early. The manuscript is well-organized and easy to follow. The novel approach of using CRP trends, rather than absolute values, to predict anastomotic leak is a potentially practice-changing finding. However, the manuscript can often be difficult to follow as the narrative is not clear. The authors present a significant amount of data without focusing on developing a clear story.

Major revisions:

  1. The introduction includes relevant research, but would benefit from explicitly describing how early detection of anastomotic leak can reduce overall morbidity, improve adjuvant chemotherapy adherence, etc. For example, the reference to the study by Gozalichvili (misspelled in manuscript) et al should describe how early detection of anastomotic leak led to greater adherence to adjuvant therapy (delayed discharge, early imaging and detection, etc)

Thank you for this response. We have expanded on this topic in our discussion.

2. It would improve the narrative of the study for the authors to describe why trend in CRP was chosen instead of absolute value. This question is not addressed until the results section, where it seems the authors formed their hypothesis.

Thank you for this recommendation. In the introduction section of the manuscript, we highlight the systematic review and meta-analysis by Bona and colleagues that outlined a variety in ranges of absolute CRP thresholds to diagnose postoperative AL with poor sensitivity and specificity. Based on these findings, and a perceived lack of studies looking at the trended values as a marker, we decided to evaluate both the previously studied point-value of CRP, as well as a relatively underutilized measure of trending CRP in the detection of the AL. More nuance regarding this methodology and decision-making was described in the methods section. We have provided a clearer thesis for the study in the last paragraph of the introduction to address this. 

3.It seems that the control would be more robust to include all patients that did not develop leak. Is there a reason why the authors did not keep patients for 3 total CRP values, or to have them send off a CRP value at home after discharge for a total of 3 values?

The reason we did not include patients with two or fewer postoperative CRP values is we cannot evaluate a trend without at least three values. In regard to the question of keeping patients for a total of three CRP values, the retrospective nature of this study makes it impossible for us to control that decision. The idea of an outpatient postoperative CRP value would be a wonderful addition to a prospective study to validate our findings.

4.This study only had 9 cases out of 226 patients. Do the authors believe this is enough to have a statistically significant and meaningful report? If so, can the authors please expound on how 9 cases are sufficient to describe an impactful finding? For example, it is very surprising to see that none of the patients who leaked had an operation for IBD, patients who are known to be in a chronically inflammatory state and likely on immunosuppression. It would therefore be difficult to extrapolate these findings as patients with IBD are high. The authors address the low case number in the limitations, but the authors need to include how statistical conclusions can still be drawn from such a small number.

The incidence of anastomotic leak is low which translates to slow accrual of anastomotic leak patients. The ideal research design would be a multi-institutional study to allow a larger sample size in a reasonable amount of time. However, given we are investigating a new strategy by which to utilize CRP values (trends as opposed to absolute values), we feel our data provide essential premises to support a larger, multi-institutional study. Our data show that although the sample size is small, we can still see a statistically significant difference in postoperative trends of patients with anastomotic leaks compared to controls. The purpose of the factors in Table 1 is to assess potential patient factors that could impact CRP levels. We certainly agree that larger studies are needed to confirm our findings in subpopulations. We have expanded on this limitation in our limitation section.

5.How do the authors account for patients who might have leaked after discharge and presented to an outside hospital?

Patients’ postoperative courses were followed for 90 days to evaluate for anastomotic leak. We have added this to the methods section. This did include review of notes from outside facilities that are still within our electronic medical record as well as review of follow-up notes with our providers. Our providers detail any reports of admission to facilities outside of our records in their notes.

6.Can the authors explain why nutritional status or index versus repeat operation were not included as independent variables? Repeat operations have been found to be associated with higher risk of leak among patients with IBD

The variables we chose to include were selected because they could potentially increase the variability in CRP levels. The purpose of this study is evaluating CRP trends as a tool for leak identification. Unless the risk factor for anastomotic leak could also impact CRP levels (chronic inflammatory conditions like IBD), we did not think it was relevant for this initial study. The CRP trend reflects the acute inflammatory process associated with an anastomotic leak, not the reason for the leak itself.

7.How was the decision made for re-operation on these patients?

All but one AL patient underwent re-operation after diagnosis of anastomotic leak. This was based on clinic judgment at the time of diagnosis including the degree of contamination suspected and the patient’s overall condition. We decided to not focus on this in the current report as our primary goal was to evaluate the ability of CRP trends to predict anastomotic leak. We agree this should be further examined in future larger studies to assess the clinical impact (reoperation versus medical management) of anastomotic leak diagnosis with CRP trends.

Minor revisions:

1.Can the authors include diverting ostomy as a variable in Table 1?

Yes, we have now added diverting ostomy to Table 1.

2.The graphs in Figure 1 are very confusing. There is a grey rectangle without explanation under graph B.

We are unsure if what you are referring to here is the figure legend. If so, this defines that the black curve in graphs B and D correspond with the control patients and the pink curve corresponds with leak patients.  Please clarify if there is a different rectangle you are seeing. 

  • The caption is difficult to follow with the reference letter at the end of the sentence instead of the beginning.
    • We have changed the reference letter to be before the sentence.
  • One of the case patients has a significant decline in CRP.
    • Yes, this is a critical point to our study. Although a handful of patients did experience a similar trend in their postoperative CRP values, it is important to recognize and appreciate that not every patient will strictly conform to the same trend rate. This may be explained by baseline patient characteristics (age, gender, smoking status, BMI), presence of inflammatory conditions (SLE, IBD), or specific factors inherent to their surgical course. Despite this, the patients did experience the similar early increase in CRP values by >10% as described as our marker for AL in the manuscript.
  • E references the percent decline of CRP on the first day, which is not clear. Was there a reference CRP done pre-op, or is this the percent decline from POD1 to POD2? When were labs drawn?

Thank you for this question. We do not routinely draw preoperative CRP values for our patients. The way that we have described the line you are mentioning was unnecessarily difficult to understand, and I have addressed the line as follows: “The initial degree of CRP reduction in control patients varied significantly, with a median decrease of 27.2%.” In translation, we were assessing the magnitude of decline for each individual patient at the point in which their CRP values BEGAN to decline during their course. This occurred at various points depending on the individual being evaluated.

3.In Table 2, can the authors clarify within the table itself? Making the language more straightforward will help the readers follow along. For example, 10% increase in CRP from POD2 to POD3

Thank you. We have toned down the wordiness of the table itself and made edits to make the descriptors clearer for the reader.

4.Why does anastomotic leak not include peri-anastomotic abscess that warrants a drain or antibiotics, similar to Gozalichvili's study? The author's definition only includes those who require re-operation

We chose not to include patients who might have imaging abnormalities such as perianastomotic air, phlegmon, or abscess unless we confirmed true extravasation of contrast or enteric contents.  While we certainly respect that a perianastomotic abscess is on the spectrum of anastomotic leak (i.e. microleak) we set the threshold in our study for leak as either enteric contents in the patient’s drain or within the peritoneal cavity at time of reoperation.  This was to ensure all our patients in the AL group truly had a clinically significant anastomotic leak, as those with a microleak may have a different inflammatory process, thus influencing the postoperative trend. We do agree these should be included in prospective trials as it may be the sensitivity/specificity of CRP trends differs in the case of microleaks compared to large leaks.

Our definition did not only include reoperation. We can see how the wording may be interrupted as such though. We have edited this section of the methods to make it more transparent. 

5.Instead of table 3, could the authors include blood loss, OR time, and whether a second procedure was performed in Table 1? Would be more clinically relevant.

Thank you for this recommendation. We have moved the presence of concomitant operation into the data for Table 1 and moved the Table 3 to our supplementary section of the manuscript. We did not collect data regarding EBL or OR time for this manuscript.

6.According to Figure 2, it seems that patients undergoing minimally invasive surgery are more likely to leak earlier in their post op course than those undergoing open surgery. Is this true?

We would not make that conclusion based on this data set. The three AL patients that underwent minimally invasive surgery did have their leak diagnosed early. However, if our proposed 10% increase had been used as a prompt to further investigate for an AL, some of the patients in the open group would have been diagnosed earlier. Given our results, we believe a prospective study is warranted to further define the ability of CRP trends to identify AL early.

7.Can the authors recommend a post-operative protocol for surgical teams to follow for early detection of anastomotic leak after elective colorectal surgery?

Yes, we have added a recommended protocol to our discussion.

Reviewer 2 Report

Comments and Suggestions for Authors

The authors state that monitoring CRP trends rather than absolute values is more effective for the early detection of anastomotic leaks (AL) after colorectal surgery. In particular, a CRP increase of 10% or more after postoperative day 2 demonstrated 100% sensitivity and 84% specificity for AL detection, showing superior diagnostic performance compared to conventional cut-off values. They also suggest that this approach could be useful for discharge decision-making in regions with limited healthcare access.

  1. The reviewer would like to ask whether differences in the incidence of anastomotic leak (AL) were examined based on the anatomical location of the anastomosis (e.g., colon vs. rectum, proximal vs. distal).
  2. The reviewer would like to point out that the number of AL patients is minimal (n=9), which may result in insufficient statistical power. Was a priori sample size calculation performed to ensure the ability to detect statistically significant differences?
  3. The reviewer would like to ask for a clear justification for selecting a 10% increase in CRP as the threshold for diagnosing AL. Why was this specific percentage chosen, and how does it compare to other possible thresholds (e.g., 5%, 15%)? Was any empirical or physiological basis considered in the selection?
  4. The reviewer acknowledges that the study demonstrates the usefulness of CRP trends in predicting AL. However, no comparisons have been made with other inflammatory markers such as procalcitonin or IL-6, which are also known to play a role in detecting postoperative infections and complications. Additionally, while CRP trends showed significant differences between AL and control groups, WBC trends did not. Why do you think WBC failed to show a similar predictive trend? The reviewer requests a discussion on these points, including potential advantages or limitations of CRP compared to other biomarkers.

Author Response

Reviewer #2

The authors state that monitoring CRP trends rather than absolute values is more effective for the early detection of anastomotic leaks (AL) after colorectal surgery. In particular, a CRP increase of 10% or more after postoperative day 2 demonstrated 100% sensitivity and 84% specificity for AL detection, showing superior diagnostic performance compared to conventional cut-off values. They also suggest that this approach could be useful for discharge decision-making in regions with limited healthcare access.

  1. The reviewer would like to ask whether differences in the incidence of anastomotic leak (AL) were examined based on the anatomical location of the anastomosis (e.g., colon vs. rectum, proximal vs. distal).

Yes, this question was addressed in Table 1. We were able to delineate the rates of AL based on type of primary anastomosis and location of said anastomosis. These options included: small bowel-small bowel, ileocolic, ileorectal, colo-colonic, colorectal (low-mid-high), ileoanal, coloanal, and diverting ostomy.

  1. The reviewer would like to point out that the number of AL patients is minimal (n=9), which may result in insufficient statistical power. Was a priori sample size calculation performed to ensure the ability to detect statistically significant differences?

We agree the AL sample size is small. We did not perform priori sample size calculations. However, given the significant differences we observed, we were concerned more for the presence of a confounding variable. For this reason, we collected data on potential confounding variables and showed no difference between groups in Table 1.

  1. The reviewer would like to ask for a clear justification for selecting a 10% increase in CRP as the threshold for diagnosing AL. Why was this specific percentage chosen, and how does it compare to other possible thresholds (e.g., 5%, 15%)? Was any empirical or physiological basis considered in the selection?

Typically, we would have used a receiver operating curve to find the best threshold. However, this is complicated when looking at trends over time. Each patient has multiple values for day-to-day changes (Ex. day 2 to 3, day 3 to 4, etc). Therefore, we instead utilized the data in figure 1E to guide our decision. Ten percent was the lower end of initial decline seen in control patients. We calculated sensitivity, specificity, and predictive values for any 10% decline seen between postoperative days as well as 10% increase between postoperative days. We found the 10% increase was better at diagnosing a leak than the 10% decrease was at ruling out a leak.

  1. The reviewer acknowledges that the study demonstrates the usefulness of CRP trends in predicting AL. However, no comparisons have been made with other inflammatory markers such as procalcitonin or IL-6, which are also known to play a role in detecting postoperative infections and complications. Additionally, while CRP trends showed significant differences between AL and control groups, WBC trends did not. Why do you think WBC failed to show a similar predictive trend? The reviewer requests a discussion on these points, including potential advantages or limitations of CRP compared to other biomarkers.

As you acknowledged, IL-6 and procalcitonin, like CRP, may aid in the diagnosis and detection of pro-inflammatory states such as AL. Several studies have attempted to validate the utility of these biomarkers, with fairly mixed results. We did not include these markers simply because they are not routinely obtained on our patients and this was a retrospective study. We do expand on the role of these markers now in our discussion. We also discuss our hypothesis regarding the lack of difference seen in WBC trends.

Round 2

Reviewer 2 Report

Comments and Suggestions for Authors

The reviewer believes the authors have adequately and appropriately responded to the reviewers’ comments and concerns, respectively.